# Seizure onset and offset pattern determine the entrainment of the cortex and substantia nigra in the nonhuman primate model of focal temporal lobe seizures

**Mark J. Connolly**[1,2], **Sujin Jiang**[3], **Lim C. Samuel**[3], **Claire-Anne Gutekunst**[4], **Robert E. Gross**[2,4,5], **Annaelle Devergnas**[1,6]*

1 Emory National Primate Research Center, Emory University, Atlanta, GA, United States of America, 2 Wallace H. Coulter Department of Biomedical Engineering, Emory University and Georgia Institute of Technology, Atlanta, GA, United States of America, 3 Emory College of Arts & Sciences, Emory University, Atlanta, GA, United States of America, 4 Department of Neurosurgery, Emory University School of Medicine, Atlanta, GA, United States of America, 5 Department of Neurosurgery, Robert Wood Johnson Medical School, Rutgers University, New Brunswick, NJ, United States of America, 6 Department of Neurology, Emory University School of Medicine, Atlanta, GA, United States of America

* adeverg@emory.edu

**Data Availability Statement:** Original data are available on the public Dandi repository: Devergnas, Annaelle (2024) SN recording during

## Abstract

Temporal lobe epilepsy (TLE) is the most common form of drug-resistant epilepsy. A major focus of human and animal studies on TLE network has been the limbic circuit. However, there is also evidence suggesting an active role of the basal ganglia in the propagation and control of temporal lobe seizures. Here, we characterize the involvement of the substantia nigra (SN) and somatosensory cortex (SI) during temporal lobe (TL) seizures induced by penicillin injection in the hippocampus (HPC) of two nonhuman primates. The seizure onset and offset patterns were manually classified and spectral power and coherence were calculated. We then compared the 3-second segments recorded in pre-ictal, onset, offset and post-ictal periods based on the seizure onset and offset patterns. Our results demonstrated an involvement of the SN and SI dependent on the seizure onset and offset pattern. We found that low amplitude fast activity (LAF) and high amplitude slow activity (HAS) onset patterns were associated with an increase in activity of the SN while the change in activity was limited to LAF seizures in the SI. However, the increase in HPC/SN coherence was specific to the farther-spreading LAF onset pattern. As for the role of the SN in seizure cessation, we observed that the coherence between the HPC/SN was reduced during burst suppression (BS) compared to other termination phases. Additionally, we found that this coherence returned to normal levels after the seizure ended, with no significant difference in post-ictal periods among the three types of seizure offsets. This study constitutes the first demonstration of TL seizures entraining the SN in the primate brain. Moreover, these findings provide evidence that this entrainment is dependent on the onset and offset pattern and support the hypothesis that the SN might play a role in the maintenance and termination of some specific temporal lobe seizure.

Temporal lobe seizures (Version 0.240621.2139) [Data set]. DANDI archive. https://doi.org/10.48324/dandi.001069/0.240621.2139

**Funding:** This work was supported by The National Institutes of Health National Institute of Neurological Disorders and Stroke grant UG3-NS100559, and by the National Institutes of Health's Office of the Director, Office of Research Infrastructure Programs P51 OD011132.

**Competing interests:** We have read the journal's policy and the authors of this manuscript have the following competing interests:Robert E. Gross serves as a consultant to Medtronic, which manufactures products related to the research described in this manuscript and receives compensation for these services. He also receives support for unrelated research. The terms of this arrangement have been reviewed and approved by Emory University in accordance with its conflict of interest policiesThis does not alter our adherence to PLOS ONE policies on sharing data and materials.

**Abbreviations:** TLE, Temporal lobe epilepsy; SN, substantia nigra; SI, somatosensory cortex; TL, temporal lobe; HPC, hippocampus; LAF, low amplitude fast activity; HAS, high amplitude slow activity; BS, burst suppression; ARR, arrhythmic offset pattern; RHY, rhythmic offset pattern; BG, basal ganglia; LFP, local field potential; ECoG, electrocorticography.

# 1. Introduction

Epilepsy affects up 1% of the population with temporal lobe epilepsy (TLE) being the most common form of drug-resistant partial epilepsy [1]. For patients with drug resistant TLE, therapies include surgery to resect the epileptogenic tissue [2] or deep brain stimulation either at the focus (e.g., hippocampus) or within the same circuit (e.g., anterior nucleus of the thalamus [3–5]. While effective at reducing seizures, these therapies do not often result in complete seizure-freedom [6]. One way to improve treatments for TLE may be to account for the propagation of the seizure to other extralimbic structures. A major focus of human and animal studies on TLE network has been the limbic circuit and the structures composing the temporal lobe. However, there is also evidence suggesting an active role of the basal ganglia (BG) in the propagation and control of temporal lobe seizures. The BG network is a complex grouping of interconnected structures that receive diverse and topographically organized inputs from the cortex and thalamus [7, 8] and is indirectly connected to the hippocampus (HPC), and amygdala [9–12]. Studies using intracranial recording have reported changes in oscillatory activity of BG structures including the putamen, pallidum, and caudate when temporal lobe seizures propagated beyond the temporal lobe [13, 14] and in the putamen during temporal lobe seizures with ictal limb dystonia [15]. In addition, a PET study in patients with temporal and extratemporal epilepsy found a decreased uptake of [¹⁸F] fluoro-L-dopa in the substantia nigra (SN), caudate, and putamen compared to healthy controls [16]. As one of the major outputs of the BG, the SN has been proposed to be a critical node responsible for the maintenance of temporal lobe seizures. In rodents, direct and indirect inhibition of the SN results in control of amygdala-kindled seizures [17–19]. Moreover, inhibition of the SN, particularly the anterior SN has been shown to be anti-ictogenic across a range of different models of focal and generalized seizures [20–26]. Similarly, low frequency stimulation of the SN has been found to suppress seizures in a cat penicillin (PCN) hippocampal seizure model [27]. Likewise, it has been shown that temporal lobe (TL) seizures can modulate activity in frontal cortical areas [28]. Studies have also consistently shown cognitive decline in patients with TLE [29]. Finally, TL seizures are associated with reduced connectivity between the temporal lobe and other cortical structures including the sensorimotor cortex (SI) [30]. Evidence also suggests that the network involved in temporal lobe seizure propagation may depend on the seizure dynamics. Most studies have focused their interest on seizure onset pattern [31–35] yet, studies on seizure termination may also provide valuable information to develop new personalized treatments to improve seizure outcome [36, 37]. So far, studies on the relationship between extralimbic activity and the onset and offset dynamic of temporal lobe seizures have been limited [35]. Here, we characterize the involvement of the SN and SI during temporal lobe seizures induced in two nonhuman primates (NHP) and test the hypothesis that seizure differentially propagate through these neural circuits based on their onset and offset pattern.

# 2. Materials and methods

## 2.1 Overview

We recorded local field potential (LFP) activity from the HPC and SN in two NHPs during temporal lobe seizures induced by local injection of penicillin in the HPC. Using these recordings, we characterized the changes in the oscillatory activity of the SN and SI during the ictal period and how the coherence between the HPC and SN or SI changes depending on the seizure onset and offset pattern.

## 2.2 Animals, surgical procedures

Two rhesus macaques (macaca mulatta; 6–10 kg; one male, one female) were used for electrophysiological recording of the HPC and SN during temporal lobe seizures. The animals were pair-housed with other animals and had free access to food and water. All experiments were performed in accordance with the United States Public Health Service Policy on the humane care and use of laboratory animals, including the provisions of the "Guide for the Care and Use of Laboratory Animals" [38]. All studies were approved by the Institutional Bio-safety and Animal Care and Use Committees (IACUC) of Emory University. The animals were first habituated to be handled by an experimenter and to sit in a primate chair. They then underwent aseptic surgery under isoflurane anesthesia (1–3%). Both animals had a recording chamber (Crist Instruments, Hagerstown, MD; inner chamber diameter 18 mm) implanted on their right side. The chamber was placed vertically in the coronal plane to permit simultaneous access to the HPC (for recording and injection) and the SN (for recording) (Fig 1B). NHP 2 had an additional electrode lead chronically implanted in the HPC (D08-15AM, 2 mm contact length, 1.5 mm between contacts, DIXI medical, France) and 2 ECoG electrodes located in the ipsilateral SI.

## 2.3 Seizure induction

Seizures were induced by an intrahippocampal injection of PCN diluted in sterile water (Penicillin G sodium salt (P3032), and sterile water for injection from Sigma-Aldrich, St. Louis, MI). Final concentration of PCN was 1,000 I.U./µl and for each session a dose of 6,000–10,000 I.U. of PCN was delivered at a rate of 1 µl/min using an injection system connected to a mechanical pump and microsyringe (CMA, Harvard Biosciences Inc, Krista, Sweden). At the end of the injection, the system was clamped to prevent any additional PCN leakage and carefully raised. Details of the model have been previously described [39]. Each PCN injection induced multiple spontaneous self-terminating seizures over a period of 4–6 hours and injections were separated by at least 2 weeks. No spontaneous behavioral seizures were observed in between injections. The seizures induced were focal and not associated with obvious clinical motor symptoms. No oral automatisms were noted during ictal events.

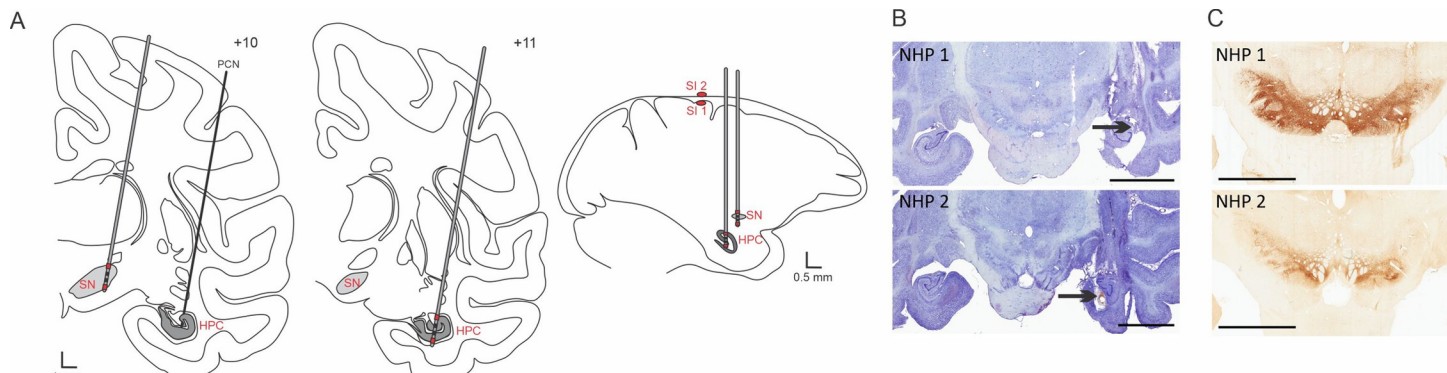

**Fig 1. Electrode targeting.** (A) Schematic representation of the electrode placement in the SN and PCN injection site in the HPC as well as electrode placement in the HPC and SI. The signal from the red contacts located in the ventral and dorsal part of the structure and in SI were subtracted from each other to obtain a bipolar recording signal in each structure. (B) Nissl stained sections showing the PCN injection site (black arrow) and electrode insertion tracts in the HPC for NHP 1 and 2. (C) TH staining showing the electrode tract in the SN for NHP 1 and 2. Abbreviations: Substantia nigra pars reticulata (SN), Hippocampus (HPC), Penicillin (PCN). Scale bars: B: 9 mm, C: 6 mm.

## 2.4 Electrophysiological mapping and recording

Recordings were made while the animals were seated in a primate chair with their head fixed but body and limbs were free to move. Neural activity was recorded using a Cerebus data acquisition system (Blackrock Neurotech, Salt Lake City, UT). A post-surgery MRI was performed to orient the chamber and the locations of SN and HPC were confirmed by extracellular activity recorded with tungsten microelectrodes (FHC, Bowdoinham, ME; Z = 0.5–1.0MΩ). The dura was pierced with a guide cannula and the electrode lowered into the brain with a microdrive (Nan instrument, Nof Hagalil, Israel). After delineating the SN and HPC borders, both structures were targeted for simultaneous LFP recordings with a 12-contact electrode consisting of 4 rings divided in 3 segments (outer diameter 600μm; inter-contact separation, 500μm; impedance, 25–35 kΩ; Heraeus Medical Components, St Paul, MN). On each recording day, one electrode was placed in the SN and the other in the HPC for simultaneous recordings from the two structures. The injection system was then lowered into the HPC, and seizures were induced. For NHP 2, electrocorticography (ECoG) from the SI was simultaneously recorded using a bipolar montage. LFP and ECoG signals were sampled at 2 kHz synchronized with the video of the animal.

## 2.5 Data processing and analysis

**2.5.1 Pre-processing.** Data were imported in Spike 2 interface (CED, Cambridge, UK) for offline annotation of ictal activity. Ictal activities were manually marked based on electrophysiological signal. We recorded 14 and 59 seizures respectively in NHP 1 and NHP 2. Subsequent analysis was performed using MATLAB (MATLAB R2021a, The Mathworks, Natick, MA, USA). The signal from each electrode contact was first normalized, then the 3 directional contacts positioned at the same level were averaged to mimic a configuration analogous to a clinical concentric 4x1 electrode. Finally, the signal from the contact located in the ventral part of the structure was subtracted from the signal in the dorsal part to obtain a bipolar recording signal in each structure (Fig 1A).

**2.5.2 Seizure classification.** The seizure onset and offset patterns were classified by a trained experimenter based on the three first and last seconds of the seizure.

Onset pattern: We identified 2 common seizure onset patterns. The low amplitude fast (LAF) activity onset pattern is characterized by low amplitude oscillations in the beta range that increases in amplitude as the seizure progresses. The high amplitude slow (HAS) activity onset pattern is characterized by high amplitude low frequency oscillations at seizure onset [32, 40].

Offset pattern: We identified 3 characteristic seizure offset patterns. The arrhythmic offset pattern (ARR) is characterized by spike of irregular amplitudes, and mixed and oscillatory frequencies. The rhythmic offset pattern (RHY) is characterized by high amplitude spikes and consistent inter-spike intervals. The burst suppression offset pattern (BS) is characterized by high amplitude, high frequency bursts interrupted by a period of suppressed activity [36, 37, 41].

The distribution of seizure onset and offset patterns was compared between animals using a Chi-square test.

**2.5.3 Analysis.** Simultaneous recordings from the HPC, SN, and SI were analyzed using the Chronux analysis software [42]. The individual power spectrogram and coherogram between signals was calculated using the multi-taper time-frequency method (moving window of 1 second and a window shift of 0.25 seconds). We extracted the power spectral density from the 3 seconds of signal preceding the seizure (pre-ictal), the 3 first seconds of the seizure onset and in the 3 last seconds of the seizure offset. The change in power and coherence were analyzed in 3 frequency bands: 1–7 Hz, 8–12 Hz and 13–25 Hz. These frequency bands were chosen based on the spectral activity of our PCN-induced seizure and previous clinical studies

showing a cut off between LAF and HAS at 8 Hz [43]. All quantitative data were expressed as mean ± SEM. The statistical reliability of the differences between periods of interest (pre-ictal, onset, offset and post-ictal) was assessed either using a Friedman repeated measures test for paired quantitative data and Dunnett's test for post hoc analysis or a Wilcoxon Signed Rank test for paired quantitative data. Comparisons between seizure pattern were performed with a Mann-Whitney Rank Sum test. Statistical values were corrected for multiple comparisons using the Bonferroni method.

## 2.6 Histological verification of electrode placement

After completion of the experiment, the animals were euthanized with an overdose of pento-barbital sodium (100 mg/kg, i.v.) and transcardially perfused with cold oxygenated Ringer's solution, followed by a fixative containing 4% paraformaldehyde and 0.1% glutaraldehyde in a phosphate buffer (PB) solution. After perfusion, the brains were cut coronally and session encompassing the SN were immunostained using a tyrosine hydroxylase antibody or cresyl violet to identify the position of the SN and verify the location of the electrode tract (see detail of the method in S1 File).

# 3. Results

## 3.1 Early involvement of the SN during TL seizure

We first analyzed all the seizures combined and found that for both animals, HPC activity was characterized by an increase in the 1–7 Hz, 8–12 Hz and 13–25 Hz frequency bands at seizure onset (Fig 2 and S1 Table). In the offset period, these oscillatory activities were still significantly higher in the 13–25 Hz range for both animals and in the 8–12 Hz for NHP 1 (Fig 2E). In contrast, while the power of the HPC in the low frequency band 1–7 Hz was increased at the beginning of the seizure, it returned to pre-ictal levels by the offset period (Fig 2E). No change was observed in the low frequency activity of the SN but for both animals the power in the 8–12 Hz and 13–25 Hz frequency range was increased at the seizure onset (Fig 2E). These initial increases of oscillatory activity in the SN remained significantly higher during the rest of the seizure for NHP 2 but were no longer significant during the offset period for NHP 1.

## 3.2 Increase in coherence between the HPC and SN

For both animals and all seizures combined, the increase in coherence between the HPC and SN was limited to the 13–25 Hz frequency band at the seizure onset (Fig 3B). The coherence in the other frequency ranges examined was either decreased or unchanged (S2 Table).

## 3.3 Change in HPC/SN activity depends on seizure onset pattern

We identified a total of 18 seizures with an HAS onset pattern (22% and 25% of the seizures recorded in respectively NHP 1 and NHP 2) and 44 seizures with a LAF onset pattern (57% and 61% of the seizures recorded in NHP 1 and NHP 2). Eleven seizure onsets did not fall under either the HAS or LAS pattern and were therefore removed from subsequent analysis (21% and 14% respectively for NHP 1 and NHP 2). No difference was found in the proportion of seizure onset patterns between animals (Chi-square 0.22, Fig 4A–4C).

For both HAS and LAF onset seizures, the oscillatory power in the SN was increased in the 8–12 Hz and in the 13–25 Hz range (Fig 4D and S3 Table). The increase in HPC/SN coherence in the 13–25 Hz frequency range during the first 3 seconds of the seizure was specific to the LAF onset pattern (Fig 4E). No significant increase in HPC/SN coherence was found for the HAS onset pattern. However, the pre-ictal HPC/SN coherence was greater for the HAS

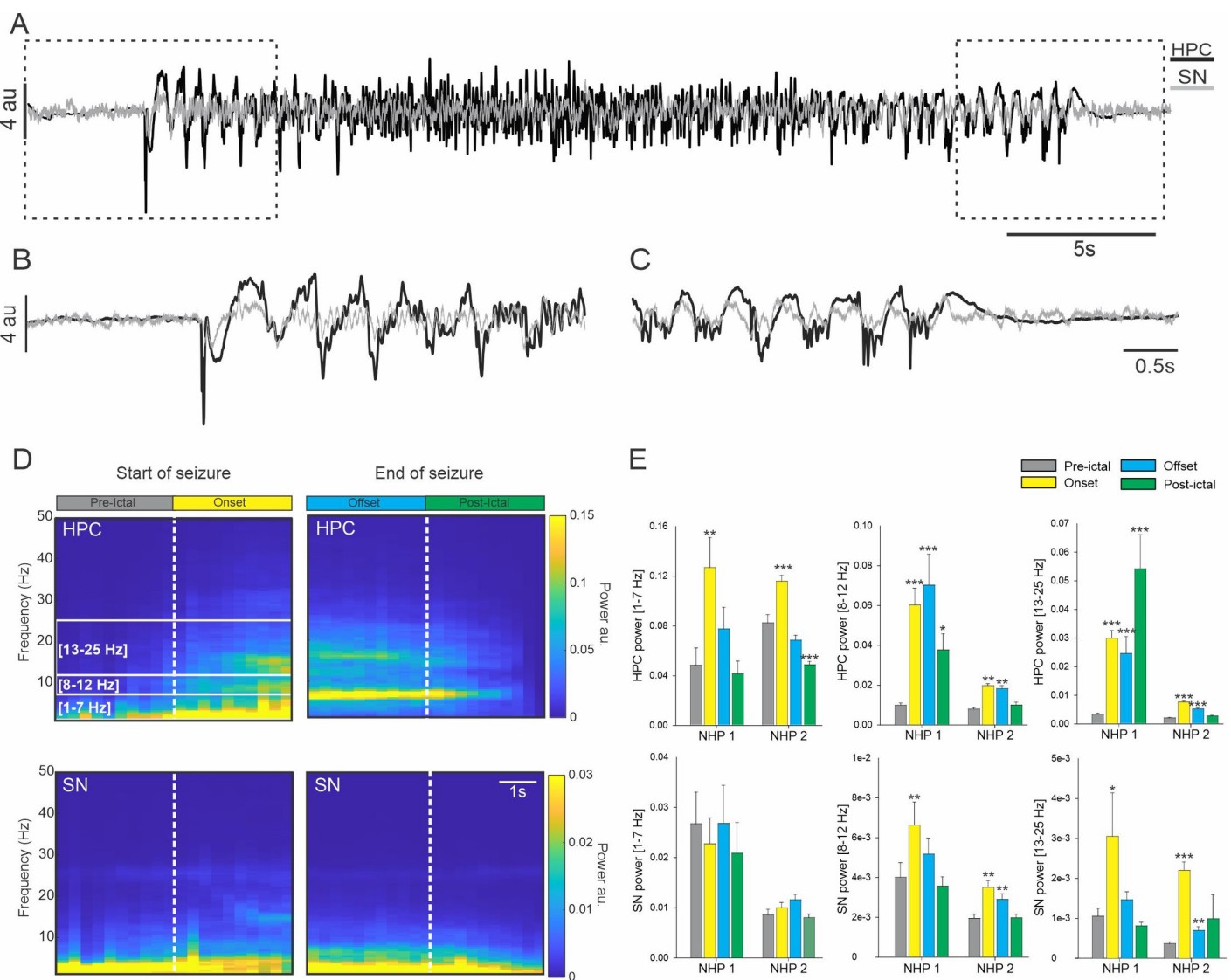

**Fig 2. Power spectral density of the SN and HPC during seizures.** (A) Example from NHP 1, of simultaneous recording obtained in the HPC and in the SN during a temporal lobe seizure with a focus on the start (B) and end (C) of the seizure. (D) Averaged spectral power obtained for NHP 1 in the HPC and SN at the start and end of the seizure. (E) Changes in the averaged power in the 1–7 Hz, 8–12 Hz and 13–25 Hz range in the HPC and SN for NHPs 1 and 2. Results are mean ± SEM. Statistical comparison performed with a Friedman repeated test and Tuckey for post hoc comparison with the pre-ictal values, *<0.05, **<0.01, ***<0.001.

seizures compared to the LAF seizures in the 1–7 Hz and 13–25 Hz range (Fig 4E and S3 Table) which might explained why we did not find an increase on the HAS seizure onset in the 13–25 Hz range.

## 3.4 Change in HPC/SN activity depends on seizure offset pattern

We identified a total of 36 seizures with an ARR offset pattern, 21 with a RHY and 12 with a BS offset pattern. Four seizure offsets did not fall under either the ARR, RHY or BS pattern and were therefore removed from subsequent analysis (Fig 5A–5D). A significant difference was found in the proportion of seizure offset patterns between animals (Fig 5D, Chi-square <0.001).

For all the offset patterns and all the frequency band analyzed, the oscillatory power in the SN was significantly decreased in the post-ictal period compared to the offset period (Fig 5E

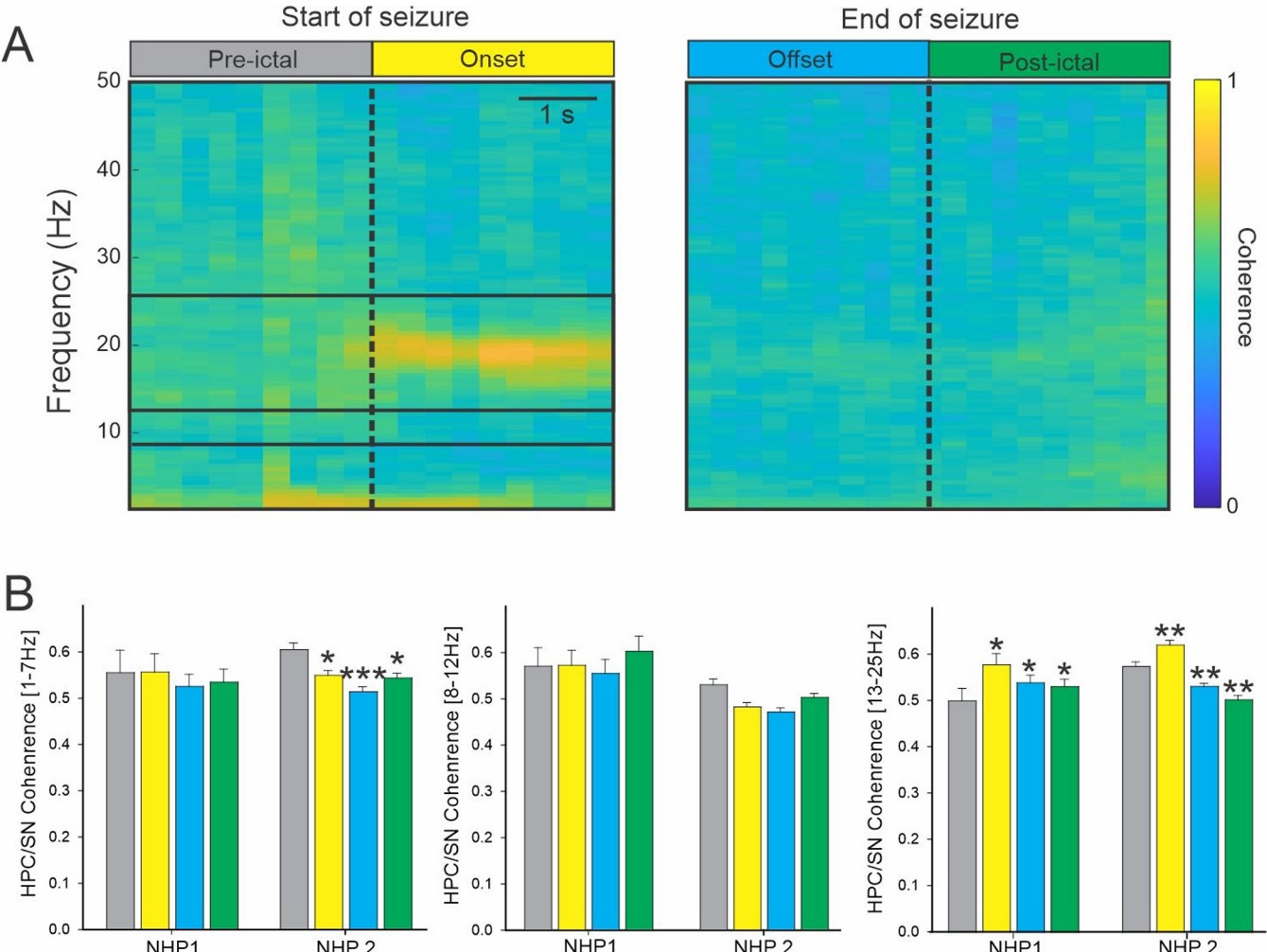

**Fig 3. Coherence between HPC and SN during seizures.** (A) Colormap representation of the averaged HPC/SN coherence obtained for NHP 2 at the start and end of the seizure. (B) The averaged HPC/SN coherence for NHP 1 and 2. Results are mean ± SEM. Statistical comparisons were performed with a Friedman repeated test and Tuckey for post hoc comparison with the values preceding the seizures, *<0.05, **<0.01, ***<0.001.

and S4 Table). For the ARR seizure offset pattern, we found an increase in HPC/SN coherence in the post-ictal period in the 8–12 Hz frequency range. We also found an increase of coherence in the post-ictal period for the BS offset in the 1–7 Hz and 13–25 Hz range, and interestingly during the offset periods the HPC/SN coherence in the BS seizure was lower compared to the ARR and RHY (Fig 5F and S4 Table). These results suggest that the SN might be involved in the ending of the BS seizures.

### 3.5 Change in HPC/SI activity depends on seizure onset and offset pattern

In NHP 2, we recorded the activity of the SI during 36 LAF and 15 HAS seizures (Fig 6A). We only found an increase of cortical oscillatory power at the beginning of the LAF seizures in the 1–7 Hz and 13–25 Hz ranges (Fig 6B and 6C and S5 Table). Interestingly, we found that for all the frequency band analyzed, the cortical activity in the pre-ictal and onset period was significantly higher during LAF compared to HAS seizures which suggest that, on the contrary to the SN, the SI is more entrained by the LAF seizure type. The coherence between the HPC and SI

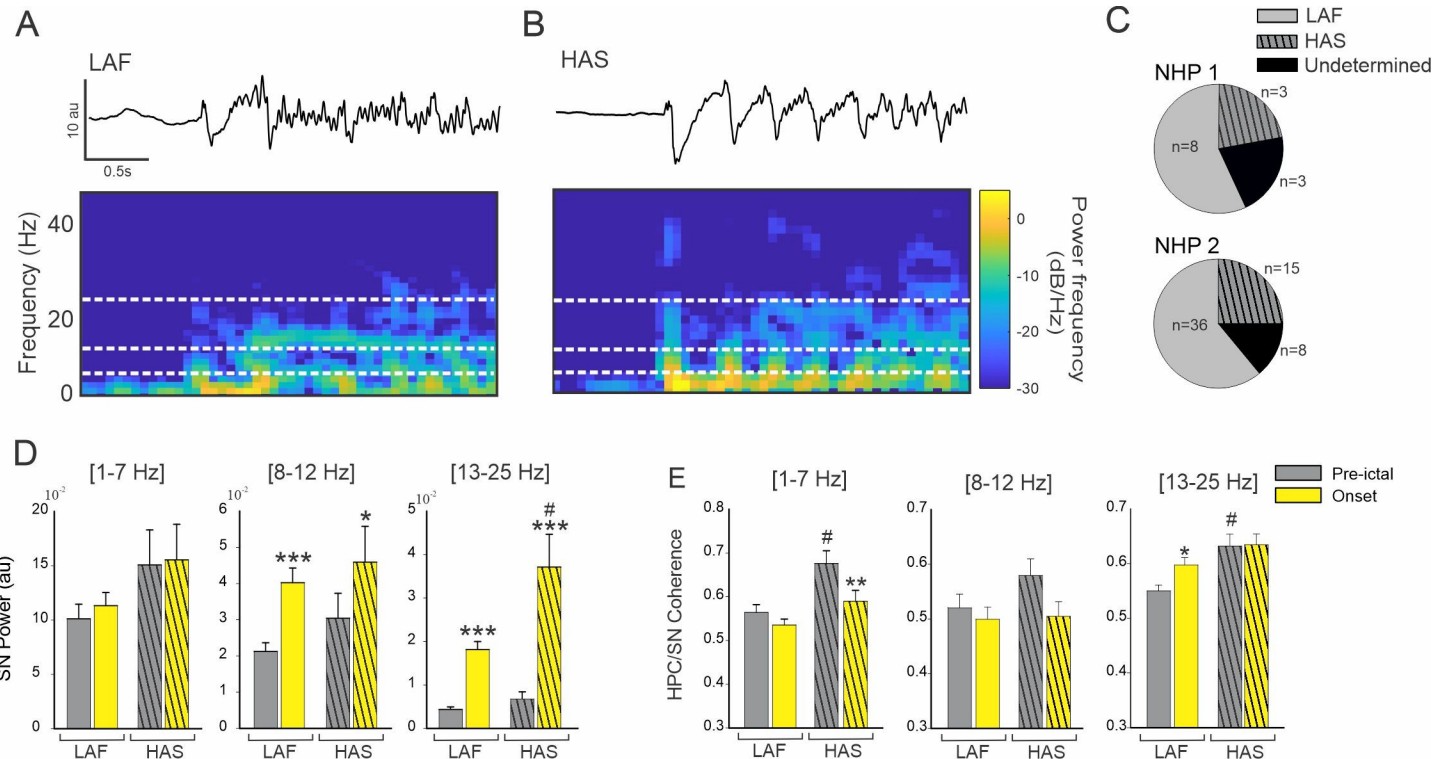

**Fig 4. Effect of seizure onset pattern on HPC/SN activity.** Example of typical LAF (A) and HAS (B) seizure showing HPC recording and corresponding spectral power. (C) Percentage of LAF, HAS, and undetermined seizures for NHP 1 and 2. (D) Changes in the averaged SN power and (E) HPC-SN coherence for LAF and HAS seizures. Results are mean ± SEM. Statistical testing for comparison of pre-ictal and onset periods was performed with a Wilcoxon signed rank test, *<0.05, **<0.01, ***<0.001. Comparisons between LAF and HAS seizures were performed with a Mann-Whitney Rank Sum test, # <0.05. Statistical values were corrected for multiple comparisons using the Bonferroni method.

was increased at the seizure onset in the 1–7 Hz range for both type of seizure and no difference in coherence was found between the LAF and HAS seizure onset. The coherence in the other frequency band were not significantly changed at the seizure onset (S5 Table).

We characterized the offset pattern of these seizures recorded in the SI and found 35 ARR, 9 RHY and 12 BS seizures offset pattern (Fig 6D and 6E and S6 Table). We found a decrease of cortical activity in the post-ictal period for the ARR and RHY pattern in the 8–12 and 13–25 Hz range (Fig 6D). In all the frequency range analyzed, cortical activity was lower during ARR seizure offset pattern compared to RHY and BS. The coherence in the 1–7 Hz range was increased in the post-ictal period of the ARR type and the HPC/SI coherence was overall higher during the RHY offset pattern. These results suggest that the SI might be more involved in the termination of ARR seizure than in the termination of RHY and BS seizure type.

## 4. Discussion

In this study we recorded LFP activity in the HPC, SN, and SI during PCN-induced temporal lobe seizures. We found a global increase of activity in SN and HPC/SN coherence which confirm the involvement of the SN during temporal lobe seizures. Based on our results, it also seems the SN and SI involvement depend on the seizure onset and offset pattern. Seizures with both LAF and HAS onset patterns were associated with an increase of activity in the SN while the change in activity was limited to LAF seizures in the SI. However, the increase of HPC/SI coherence was similar for both type of onset, while the increase in HPC/SN coherence was

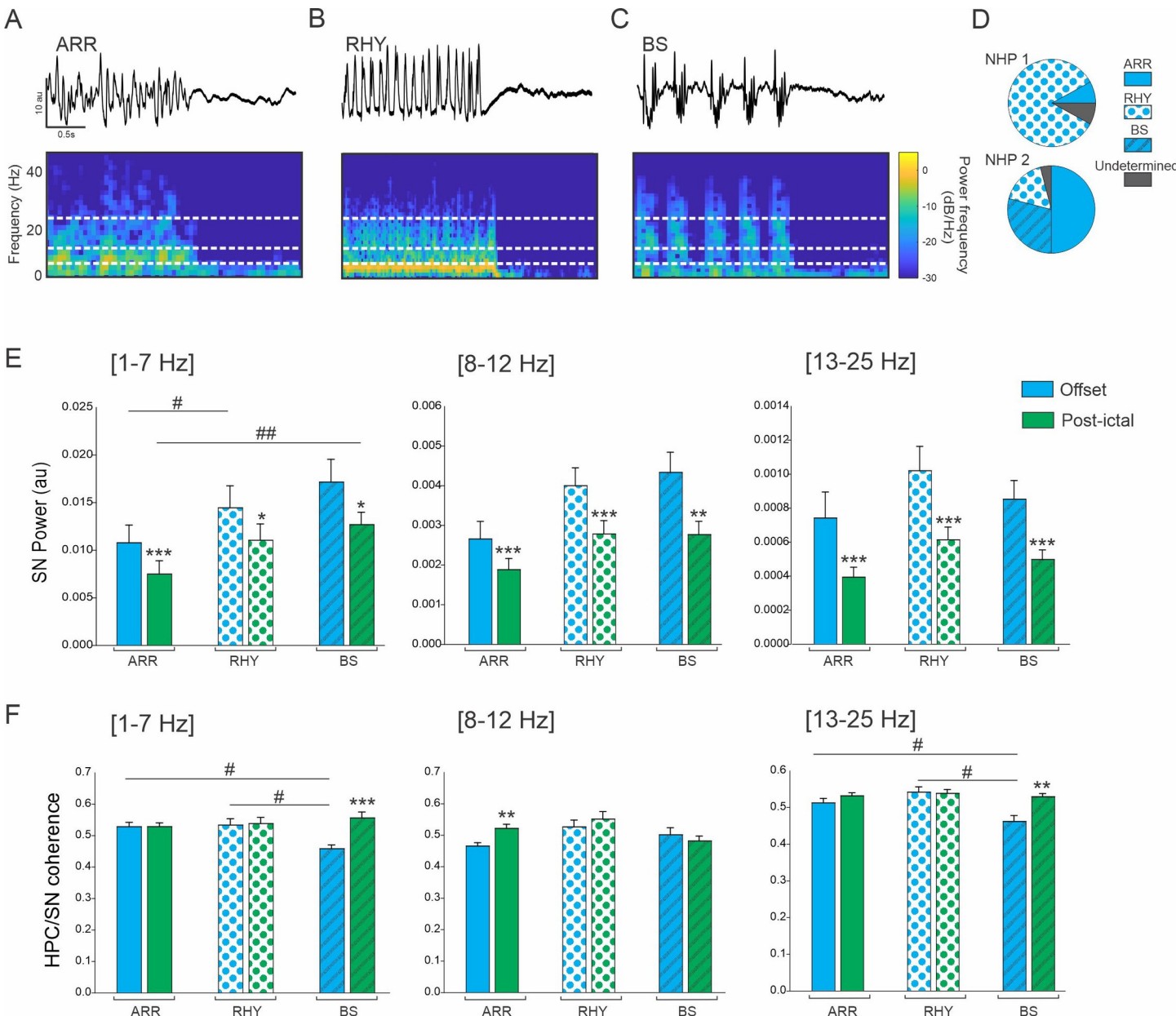

**Fig 5. Effect of seizure offset pattern on HPC/SN activity.** Example of typical ARR (A), RHY (B) and BS (C) seizure offset pattern. (D) Percentage of ARR, RHY, BS and undetermined seizures for NHP 1 and 2. (E) Changes in the averaged SN power and (F) HPC-SN coherence for LAF and HAS seizures. Results are mean ± SEM. Statistical comparison for comparison of offset and post-ictal period was performed with a Wilcoxon signed rank test, *<0.05, **<0.01, ***<0.001. Comparisons between ARR, RHY and BS seizures were performed with an ANOVA and Dunn's Post hoc test, # <0.05, ##<0.01. Statistical values were corrected for multiple comparison using the Bonferroni method.

specific to the farther-spreading LAF onset pattern. With regard to the involvement of the SN at the offset, the coherence HPC/SN was lower during BS compared to other ending and this coherence was normalized in the post-ictal period with no difference in post-ictal phase between the 3 types of seizure offset. This result suggests that the SN might be involved in the termination of the BS seizure pattern. This study is the first demonstration of temporal lobe seizures entraining the SN in the primate brain. Moreover, these findings provide evidence that this entrainment is dependent on the seizure onset pattern and support the hypothesis

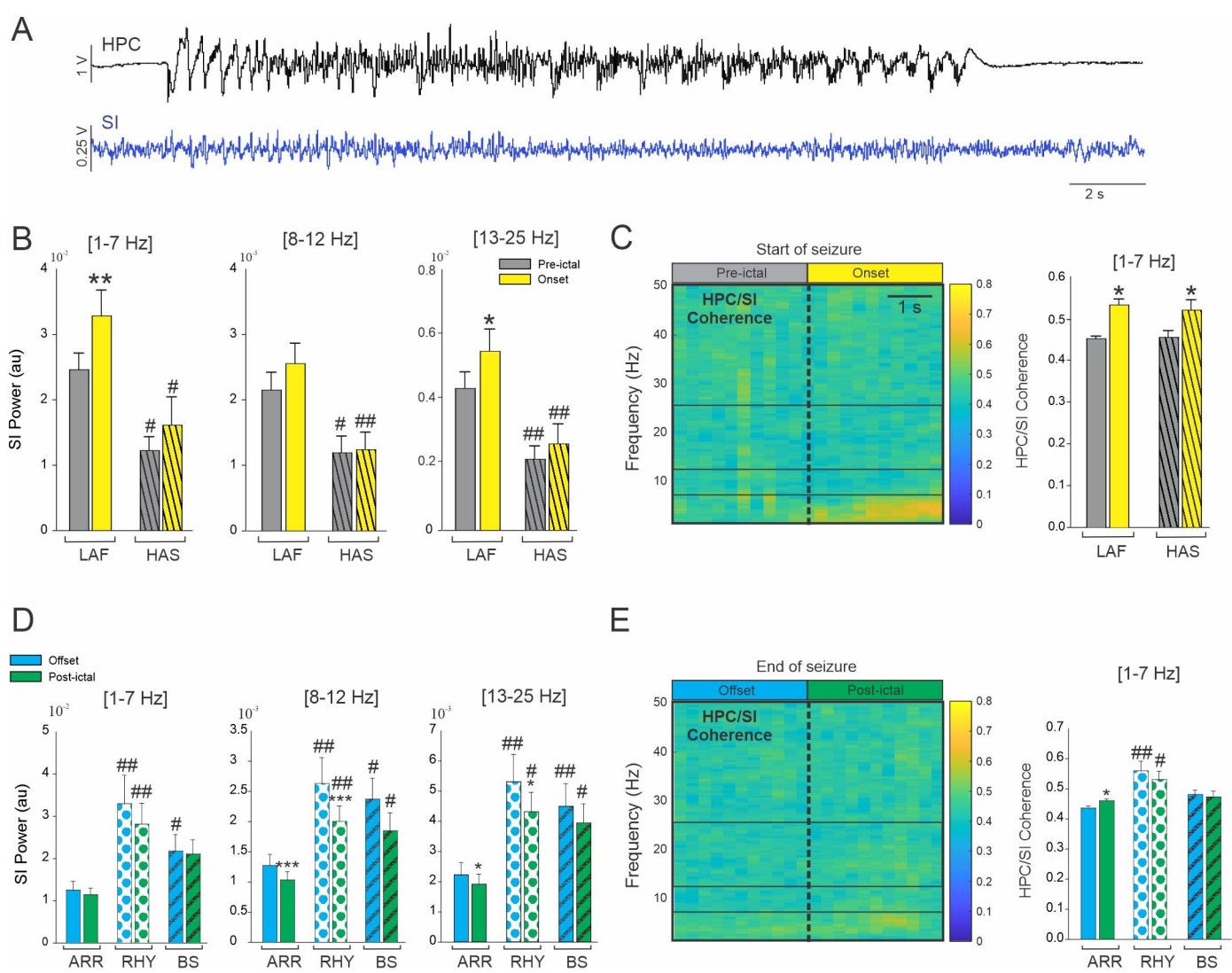

**Fig 6. Influence of temporal lobe seizure onset and offset pattern on cortical activity.** (A) Example of simultaneous recordings obtained from the HPC and the SI in NHP 2. (B) Changes in the averaged SI power for LAF and HAS seizures. (C) Colormap of the averaged HPC/SI coherence (LAF and HAS combined) and changes in the averaged HPC/SI coherence in the 1–7 Hz for LAF and HAS seizures. (D) Changes in the averaged SI power for ARR, RHY and BS seizures. (C) Colormap of the averaged HPC/SI coherence (LAF and HAS combined) and changes in the averaged HPC/SI coherence in the 1–7 Hz for LAF and HAS seizures. Results are mean ± SEM. Statistical comparisons between pre-ictal vs onset and offset vs postictal were performed with Wilcoxon signed rank test *<0.05, **<0.01 and ***<0.001. Comparisons between LAF and HAS seizures were performed with a Mann-Whitney Rank Sum test, and comparison between ARR, RHY and BS were performed with an ANOVA on rank and Dunn's Post hoc test # <0.05, ##<0.01. Statistical values were corrected for multiple comparison using the Bonferroni method.

that the SN might plays a role in the maintenance and termination of specific temporal lobe seizures.

## 4.1 The role of the SN in the control of seizures

As one of the major outputs of the BG, the SN has been proposed to be a critical node responsible for the maintenance of temporal lobe seizures. In rodents, direct and indirect inhibition of the SN results in control of amygdala-kindled seizures [17–19]. Likewise, inhibition of the SN, particularly the anterior SN has been shown to be anti-ictogenic across a range of different models of focal and generalized seizures [20–26]. Similarly, low frequency stimulation of the

SN has been found to suppress seizures in a cat penicillin (PCN) hippocampal seizure model [27]. There is also a long history of clinical data implicating the BG in temporal and extratemporal seizures using electrophysiological recordings [13–15], measuring cerebral metabolism [44, 45], and advanced fMRI analysis techniques [46]. To our knowledge, however, entrainment of the SN during a temporal lobe seizure has never been observed in the primate brain. Our finding that the coherence between the HPC and the SN is increased during the onset of temporal lobe seizures implicates the primate SN in TL seizures as in rodents and suggests that modulation of the non-human primate SN may have similar anti-seizure effects than in rodents. This increase of activity in the SN was also significant during the offset period for NHP 2 but did not reach significance for NHP 1. We believe that this difference between the 2 animals might be explained by the lower number of seizures for NHP 1 (14 seizures) compared to NHP 2 (59 seizures).

## 4.2 Entrainment of the SN is dependent on seizure onset pattern

Because of their importance in understanding ictogenesis and predicting treatment outcome, seizure onset patterns have recently drawn increasing attention [40, 47, 48].

The onset (1–5 seconds) of a seizure typically has a morphology consistent with one of several stereotyped onset patterns. Two of the most reported seizure onset patterns in patients with temporal lobe epilepsy are LAF and HAS onset patterns. The LAF seizure onset pattern is characterized by low amplitude oscillations in the beta to gamma range that slowly increase as the seizure progresses, whereas the HAS onset pattern consists of high amplitude slow oscillations usually below the alpha range [32, 33]. While both seizure onset patterns can arise from the same ictogenic circuit, seizures with the LAF onset pattern typically spread farther and have a larger seizure onset zone [31–34, 49]. Similar to what has been described in clinical studies, our NHP model exhibited a prevalence of LAF onset pattern [32, 48, 50]. We found that both the LAF and HAS entrained the SN, but only LAF caused an increase in HPC/SN coherence. This finding corroborates clinical studies showing that LAF seizures are typically associated with a larger seizure onset zone and spread farther than HAS seizures in both temporal and extratemporal lobe seizures [31–34, 49, 51]. The finding that only the further-spreading LAF seizures influenced the SN would support the hypothesis that the SN plays a role in the propagation and generalization of seizures. However, we also found that the HPC/SN coherence during the pre-ictal period was higher for the HAS than for the LAF seizure type. This result is intriguing and should be investigated further as coherence between the HPC and other subcortical structure might convey information on seizure occurence.

## 4.3 Entrainment of the SN is dependent on seizure offset pattern

Most studies on seizure dynamics have focused on onset pattern and propagation however, a better understanding of seizure termination may provide critical information regarding seizure mechanisms and open new approaches to treat status epilepticus [52]. Some papers have classified the seizure termination patterns in patients and the classification is very similar to the one we created for our NHP-PCN model [36, 41, 53, 54]. As previously discussed, the role of SN in the termination of seizure has been investigated in several rodent studies however no study has considered the implication of the SN in the context of different seizure termination patterns. Our results shows that the HPC/SN coherence was lower during BS compared to ARR and RHY seizures offset which might suggest that the SN might be involved differently in the termination of seizure depending on their pattern. Similarly, the thalamo-cortical synchronization has been found to vary depending on the offset pattern leading to the hypothesis that hypersynchronization might ultimately leads to the termination of a subtype of seizures [41]. However,

we cannot exclude the possibility that the difference in HPC/SN coherence obtained for the BS pattern might only reflect the difference in activity at the HPC level.

This study serves as an initial step to demonstrate the role of the substantia nigra (SN) in temporal lobe (TL) seizures. However, inhibiting the SN should be pursued further to investigate its function in terminating burst suppression (BS) offset patterns. Additionally, the implication of the other BG structures should be evaluated to better understand the role of these nuclei in the propagation and termination of TL seizures. Special attention should be given to the unique roles of the direct and indirect basal ganglia pathways. A recent study has shown separate implication of these pathways in frontal lobe seizure [55] while another one found no difference between chemogenetic inhibition of direct and indirect pathway [56]. Exploring the network involvement in function of seizure focus but also the seizure pattern is a necessary step for novel individualized therapy development.

### 4.4 Propagation network and cortical entrainment of temporal lobe seizures

Multiple studies have shown the propagation of temporal lobe seizures to the SN although the pathway, mechanism, and role of the BG therein remains unclear. It has been suggested that BG activity is modulated by seizures via cortical areas [13, 57], and that the HPC-cortico-striatal connection is the entrance pathway leading to the entrainment of the BG. In support of this theory, studies have shown a preventative effect of GABA antagonist and dopamine agonist injections in the striatum [58–60]. Our results also agree with this hypothesis of cortical recruitment, as the LAF seizure that were associated with an increase in HPC/SN coherence were also associated with an increase in HPC/SI coherence. However, while we did not find a difference in HPC/SI coherence for both type of seizure onset, the SI power during HAS seizure was significantly lower than during LAF seizure at seizure onset but also in the pre-ictal state. It is possible that while both type of seizure might propagated similarly to the SI, the indirect and direct BG pathway might be engaged differently based on the SI oscillatory activity [61]. However, we understand the limitation of this study and further simultaneous recordings of the cortex and BG during TL seizure should be performed. Such study would allow for a better understanding of the propagation of each seizure type through this network and a dynamic analysis of the changes in network causality throughout the seizure [35].

### 4.5 Limitations

Although these results demonstrate the implication of the BG in TLE and the relevance of the seizure patterns when studying pathways and mechanisms, as is often the case in NHP studies the relatively small number of animals is a limitation. The model of PCN-induced seizure model could also be seen as a limitation since it is not a chronic epilepsy model, however this model provides larger amount of on-demand seizures (which was necessary for electrophysiological study) with a variety of onset and offset pattern [39, 62]. Additionally, in our study, the recordings were obtained from the posterior part of the SN on both animals. Although it is well known that the topographical segregation of efferent and afferent connections in the SN is even more pronounced in the NHP than in the rodent [63], we were unable to characterize any regional variability in SN seizure activity because - to limit neural tissue damage caused by acutely targeting deep structures - we only obtained recordings from one SN location. Future studies can address this by recording and modulating different parts of the SN to evaluate the potential topography of seizure regulation in the NHP. Along this line, we acknowledge that without comprehensive EEG recordings, we cannot conclude that the entrainment of the SN was mediated through cortical recruitment and it is possible that the entrainment of the SN

could also be due to seizure propagation via another pathway [64]. Finally, our seizure onset and offset classification was only based on the recording done at the seizure focus and the activity in other location were not taken into account. Thus, for the offset pattern we did not evaluate if the seizure was ending synchronously or asynchronously across the brain [31]. A more detailed distinction of the seizure patterns could be made in the future but would require additional seizures and recordings site [36, 37].

## 5. Conclusion

This work demonstrates the implication of the SN in temporal lobe seizures in NHP. In addition, we showed that the SN and cortex entrainment were dependent on the seizure onset and offset pattern. These results confirm the idea that seizures arising from the same focus might involve different pathways depending on their pattern of progression and termination.

## Supporting information

**S1 File. Description of the staining method.**
(DOCX)

**S1 Table. Spectral values in SN in the pre-ictal, onset, offset and post-ictal periods.**
(DOCX)

**S2 Table. Coherence obtained the pre-ictal, onset, offset and post-ictal periods.**
(DOCX)

**S3 Table. Spectral and HP/SN coherence in the pre-ictal and onset periods for LAF and HAS onset patterns.**
(DOCX)

**S4 Table. Spectral values and HP/SN coherence obtained for ARR, RHY and BS offset patterns.**
(DOCX)

**S5 Table. Spectral and HP/SI coherence in the pre-ictal and onset periods for LAF and HAS onset patterns.**
(DOCX)

**S6 Table. Spectral values and HP/SI coherence obtained for ARR, RHY and BS offset patterns.**
(DOCX)

## Author Contributions

**Conceptualization:** Robert E. Gross, Annaelle Devergnas.

**Data curation:** Mark J. Connolly, Claire-Anne Gutekunst.

**Formal analysis:** Mark J. Connolly, Sujin Jiang, Lim C. Samuel, Claire-Anne Gutekunst, Annaelle Devergnas.

**Funding acquisition:** Claire-Anne Gutekunst, Robert E. Gross, Annaelle Devergnas.

**Investigation:** Mark J. Connolly, Claire-Anne Gutekunst, Annaelle Devergnas.

**Methodology:** Claire-Anne Gutekunst, Robert E. Gross, Annaelle Devergnas.

**Project administration:** Claire-Anne Gutekunst, Robert E. Gross, Annaelle Devergnas.

**Supervision:** Claire-Anne Gutekunst, Robert E. Gross, Annaelle Devergnas.

**Writing – original draft:** Mark J. Connolly.

**Writing – review & editing:** Mark J. Connolly, Sujin Jiang, Lim C. Samuel, Claire-Anne Gutekunst, Robert E. Gross, Annaelle Devergnas.

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
