## [Decision Letter · Decision Letter 0]

24 May 2024

PONE-D-24-17382Seizure onset and offset pattern determine the entrainment of the cortex and substantia nigra in the nonhuman primate model of focal temporal lobe seizuresPLOS ONE

Dear Dr.  Devergnas,

Thank you for submitting your manuscript to PLOS ONE. After careful consideration, we feel that it has merit but does not fully meet PLOS ONE’s publication criteria as it currently stands. Therefore, we invite you to submit a revised version of the manuscript that addresses the points raised during the review process.

We look forward to receiving your revised manuscript.

Kind regards,

Ayataka Fujimoto

Academic Editor

PLOS ONE

Journal Requirements:

   "This work was supported by The National Institutes of Health National Institute of Neurological Disorders and Stroke grant UG3-NS100559" 

   "I have read the journal's policy and the authors of this manuscript have the following competing interests:Robert E. Gross serves as a consultant to Medtronic, which manufactures products related to the research described in this manuscript and receives compensation for these services. He also receives support for unrelated research. The terms of this arrangement have been reviewed and approved by Emory University in accordance with its conflict of interest policies."

Additional Editor Comments:

Two reviewers have provided their comments. Both believe that this research is significant, so please revise the manuscript according to their feedback.

Reviewers' comments:

Reviewer's Responses to Questions

**Comments to the Author**

1. Is the manuscript technically sound, and do the data support the conclusions?

Reviewer #1: Yes

Reviewer #2: Yes

2. Has the statistical analysis been performed appropriately and rigorously? 

Reviewer #1: Yes

Reviewer #2: Yes

3. Have the authors made all data underlying the findings in their manuscript fully available?

Reviewer #1: Yes

Reviewer #2: No

4. Is the manuscript presented in an intelligible fashion and written in standard English?

Reviewer #1: Yes

Reviewer #2: Yes

5. Review Comments to the Author

Reviewer #1: Dr. Connolly et al. studied the involvement of the substantia nigra and somatosensory cortex during temporal lobe seizures induced in two nonhuman primates. They found that the involvement of the SN and SI depends on the seizure onset zone and offset pattern.

Comments (invitation: May 10, 2024, and submission: May 11, 2024)

I want to congratulate the authors’ efforts. The manuscript is well written and easy to follow. None of the following comments are criticisms.

1) Lines 159-162: Would it be possible to provide the location of the electrodes to get the signal from SI in Figure1 or provide a new figure?

2) Lines 256-259: This description differs slightly from Figure 2E, causing confusion. The part stating "These oscillatory activities decreased at the end of the seizures but were still significantly higher in the 13–25 Hz range for both animals" seems to correspond to the blue "Offset" in Figure 2E. However, the part mentioning "in the 8–12 Hz for NHP 1" appears to refer to the green "Post-ictal." Perhaps using "Offset" and "Post-ictal" would clarify the statement, as the term "end of the seizures" is somewhat ambiguous.

3) Line 263-265: What might be the reason for the difference in the duration of SN power between NHP1 and NHP2? Please address this in the Discussion section.

4) Line 343-344: While it's indeed possible that the SN might be involved in the termination of BS patterns, is the SN the cause of BS pattern termination, or are the results obtained in this study a consequence of the termination of the BS pattern? Please elaborate on the authors' thoughts in the Discussion section.

5) Line 380: The authors may want to change from “SI” to “SN”.

6) Line 398-401: What is your interpretation of the fact that only BS seizures, not ARR or RHY, are associated with the SN as observed in the results of this study? Please provide the authors' perspective. Is there a relationship between the termination pattern of BS seizures and the basal ganglia?

7) Section 4.4: In this study, the authors examined the coherence between the HPC and SI, not the relationship between SI and SN. Therefore, it may be challenging to discuss the involvement of the SN and basal ganglia in this paragraph. What are your thoughts on this?

8) Figure 2E: The authors may want to change from 8-12 Hz to 13-25 Hz in the vertical axis of the bottom-right plot.

Reviewer #2: The authors have repeatedly shown the mechanism of neural networks in nonhuman models. While the limbic circuits have been a major focus in temporal lobe epilepsy, this study shows the implication of the SN in temporal lobe seizures in NHP. I believe that this study is valuable in elucidating the circuitry of temporal lobe epilepsy.

There are a few points that could possibly improve the manuscript with a better representation of data.

1)

Why is the penicillin model being used in this study?

I understand that the purpose of this study is to investigate how the epileptic activities propagate to the other extralimbic structures in a temporal lobe epilepsy. The penicillin model is closer to a model of acute symptomatic seizures rather than chronic epilepsy. Wouldn't it be more appropriate to evaluate using a model closer to chronic epilepsy, such as kindling?

If it is convenient, that is fine. It may be mentioned in the limitation.

2)

Ictal DC shifts and ictal high-frequency oscillations (HFOs) are noted as a marker of epileptogenesis in humans.

In this study, it was mentioned that the EEG sampling rate was 2 kHz. This analysis focuses on frequencies between 1 and 25 Hz. Is it correct to understand that frequency bands above 25 Hz did not appear during seizures?

3)

It would be better to standardize the way abbreviations are noted.

In the abstract, "temporal lobe" is abbreviated as "TL," and later it appears as "TLE." Additionally, in the introduction, "hippocampus" is noted as "HPC," but in the abstract, it is immediately referred to as "HPC" without prior introduction.

It can be confusing when an abbreviation appears suddenly without prior introduction, so it would be desirable to standardize the notation method.

Creating a list of abbreviations somewhere might be a good idea.

4)

Is there any mention of data availability? If not, I would like it to be added.

6. PLOS authors have the option to publish the peer review history of their article (what does this mean?). If published, this will include your full peer review and any attached files.

Reviewer #1: No

Reviewer #2: No

---

## [Author Response · Author response to Decision Letter 0]

24 Jun 2024

Reviewer #1: 

1) Lines 159-162: Would it be possible to provide the location of the electrodes to get the signal from SI in Figure1 or provide a new figure?

Response: We added a sagittal representation on figure 1 to provide the location of the electrode placement in SI. No histological verification could be done as these electrodes were not located inside the brain. 

Figure 1: Electrode targeting. (A) Schematic representation of the electrode placement in the SN and PCN injection site in the HPC as well as electrode placement in the HPC and SI. The signal from the red contacts located in the ventral and dorsal part of the structure and in SI were subtracted from each other to obtain a bipolar recording signal in each structure. (B) Nissl stained sections showing the PCN injection site (black arrow) and electrode insertion tracts in the HPC for NHP 1 and 2. (C) TH staining showing the electrode tract in the SN for NHP 1 and 2. Abbreviations: Substantia nigra pars reticulata (SN), Hippocampus (HPC), Penicillin (PCN). Scale bars: B: 9 mm, C: 6 mm.

2) Lines 256-259: This description differs slightly from Figure 2E, causing confusion. The part stating "These oscillatory activities decreased at the end of the seizures but were still significantly higher in the 13–25 Hz range for both animals" seems to correspond to the blue "Offset" in Figure 2E. However, the part mentioning "in the 8–12 Hz for NHP 1" appears to refer to the green "Post-ictal." Perhaps using "Offset" and "Post-ictal" would clarify the statement, as the term "end of the seizures" is somewhat ambiguous.

Response: Thanks for noticing, we corrected the error and replace “end of the seizure” by “offset period” or “post-ictal” when appropriate in the revised manuscript.

3) Line 263-265: What might be the reason for the difference in the duration of SN power between NHP1 and NHP2? Please address this in the Discussion section.

Response: We added a section in the discussion to address this difference between animals.

“This increase of activity in the SN was also significant during the offset period for NHP 2 but did not reach significance for NHP 1. We believe that this difference between the 2 animals might be explained by the lower number of seizures for NHP 1 (14 seizures) compared to NHP 2 (59 seizures).” 

4) Line 343-344: While it's indeed possible that the SN might be involved in the termination of BS patterns, is the SN the cause of BS pattern termination, or are the results obtained in this study a consequence of the termination of the BS pattern? Please elaborate on the authors' thoughts in the Discussion section.

Response: Thanks, this is an interesting point, we added a section in the discussion to address it.

“However, we cannot exclude the possibility that the difference in HPC/SN coherence obtained for the BS pattern might only reflect the difference in activity at the HPC level. 

This study serves as an initial step to demonstrate the role of the substantia nigra (SN) in temporal lobe (TL) seizures. However, inhibiting the SN should be pursued further to investigate its function in terminating burst suppression (BS) offset patterns. Additionally, the implication of the other BG structures should be evaluated to better understand the role of these nuclei in the propagation and termination of TL seizures. Special attention should be given to the unique roles of the direct and indirect basal ganglia pathways. A recent study has shown separate implication of these pathways in frontal lobe seizure 1 while another one found no difference between chemogenetic inhibition of direct and indirect pathway 2. Exploring the network involvement in function of seizure focus but also the seizure pattern is a necessary step for novel individualized therapy development. “

5) Line 380: The authors may want to change from “SI” to “SN”.

Response: Thanks, this has been corrected.

6) Line 398-401: What is your interpretation of the fact that only BS seizures, not ARR or RHY, are associated with the SN as observed in the results of this study? Please provide the authors' perspective. Is there a relationship between the termination pattern of BS seizures and the basal ganglia?

Response: We added a section in the discussion to address this 

“

However, we cannot exclude the possibility that the difference in HPC/SN coherence obtained for the BS pattern might only reflect the difference in activity at the HPC level. 

This study serves as an initial step to demonstrate the role of the substantia nigra (SN) in temporal lobe (TL) seizures. However, inhibiting the SN should be pursued further to investigate its function in terminating burst suppression (BS) offset patterns. Additionally, the implication of the other BG structures should be evaluated to better understand the role of these nuclei in the propagation and termination of TL seizures. Special attention should be given to the unique roles of the direct and indirect basal ganglia pathways. A recent study has shown separate implication of these pathways in frontal lobe seizure 1 while another one found no difference between chemogenetic inhibition of direct and indirect pathway 2. Exploring the network involvement in function of seizure focus but also the seizure pattern is a necessary step for novel individualized therapy development. “

7) Section 4.4: In this study, the authors examined the coherence between the HPC and SI, not the relationship between SI and SN. Therefore, it may be challenging to discuss the involvement of the SN and basal ganglia in this paragraph. What are your thoughts on this?

Response: It has been suggested that BG involvement during TL seizure originated from the cortex however we recognize that our current SI data set does not allow for a conclusion on BG involvement. We added this in the discussion.

“However, we understand the limitation of this study and further simultaneous recordings of the cortex and BG during TL seizure should be performed. Such study would allow for a better understanding of the propagation of each seizure type through this network and a dynamic analysis of the changes in network causality throughout the seizure 3.”

8) Figure 2E: The authors may want to change from 8-12 Hz to 13-25 Hz in the vertical axis of the bottom-right plot.

Response: Thanks, this has been corrected.

Reviewer #2: The authors have repeatedly shown the mechanism of neural networks in nonhuman models. While the limbic circuits have been a major focus in temporal lobe epilepsy, this study shows the implication of the SN in temporal lobe seizures in NHP. I believe that this study is valuable in elucidating the circuitry of temporal lobe epilepsy.

There are a few points that could possibly improve the manuscript with a better representation of data.

1)

Why is the penicillin model being used in this study?

I understand that the purpose of this study is to investigate how the epileptic activities propagate to the other extralimbic structures in a temporal lobe epilepsy. The penicillin model is closer to a model of acute symptomatic seizures rather than chronic epilepsy. Wouldn't it be more appropriate to evaluate using a model closer to chronic epilepsy, such as kindling?

If it is convenient, that is fine. It may be mentioned in the limitation.

Response: The penicillin model was chosen because we needed a large number of seizures exhibiting different onset and offset pattern. Unfortunately, the NHP chronic model do not display as many seizures as the on-demand model 4. In addition, these models are more difficult to manage as most of the time a status epilepticus precede the occurrence of spontaneous recurrent seizures. We added this in our limitation.

2)

Ictal DC shifts and ictal high-frequency oscillations (HFOs) are noted as a marker of epileptogenesis in humans.

In this study, it was mentioned that the EEG sampling rate was 2 kHz. This analysis focuses on frequencies between 1 and 25 Hz. Is it correct to understand that frequency bands above 25 Hz did not appear during seizures?

Response: We intentionally concentrated our analysis on the lower frequencies as they highlighted the differences between various onset and offset patterns. Nevertheless, higher frequencies are also present and could be examined in a subsequent analysis.

3)

It would be better to standardize the way abbreviations are noted.

In the abstract, "temporal lobe" is abbreviated as "TL," and later it appears as "TLE." Additionally, in the introduction, "hippocampus" is noted as "HPC," but in the abstract, it is immediately referred to as "HPC" without prior introduction.

It can be confusing when an abbreviation appears suddenly without prior introduction, so it would be desirable to standardize the notation method.

Creating a list of abbreviations somewhere might be a good idea.

Response: Thanks for noticing. We corrected the errors in abbreviations and created a list of abbreviation.

4)

Is there any mention of data availability? If not, I would like it to be added.

Response: Original data are available on the public Dandi repository:

Devergnas, Annaelle (2024) SN recording during Temporal lobe seizures (Version 0.240621.2139) [Data set]. DANDI archive. https://doi.org/10.48324/dandi.001069/0.240621.2139

1. Brodovskaya A, Shiono S, Kapur J. Activation of the basal ganglia and indirect pathway neurons during frontal lobe seizures. Brain. Aug 17 2021;144(7):2074-2091. doi:10.1093/brain/awab119

2. Zou W, Guo Z, Suo L, et al. Nucleus accumbens shell modulates seizure propagation in a mouse temporal lobe epilepsy model. Front Cell Dev Biol. 2022;10:1031872. doi:10.3389/fcell.2022.1031872

3. Ilyas A, Toth E, Chaitanya G, Riley K, Pati S. Ictal high-frequency activity in limbic thalamic nuclei varies with electrographic seizure-onset patterns in temporal lobe epilepsy. Clin Neurophysiol. May 2022;137:183-192. doi:10.1016/j.clinph.2022.01.134

4. Croll L, Szabo CA, Abou-Madi N, Devinsky O. Epilepsy in nonhuman primates. Epilepsia. Aug 2019;60(8):1526-1538. doi:10.1111/epi.16089

---

## [Decision Letter · Decision Letter 1]

15 Jul 2024

Seizure onset and offset pattern determine the entrainment of the cortex and substantia nigra in the nonhuman primate model of focal temporal lobe seizures

PONE-D-24-17382R1

Dear Dr. Annaelle delphine Devergnas,

We’re pleased to inform you that your manuscript has been judged scientifically suitable for publication and will be formally accepted for publication once it meets all outstanding technical requirements.

Kind regards,

Ayataka Fujimoto

Academic Editor

PLOS ONE

Additional Editor Comments (optional):

Two reviewers have decided to accept the manuscript. The authors have sincerely addressed the reviewers' comments, and I thank them for that. I will also accept it.

Reviewers' comments:

Reviewer's Responses to Questions

**Comments to the Author**

1. If the authors have adequately addressed your comments raised in a previous round of review and you feel that this manuscript is now acceptable for publication, you may indicate that here to bypass the “Comments to the Author” section, enter your conflict of interest statement in the “Confidential to Editor” section, and submit your "Accept" recommendation.

Reviewer #1: All comments have been addressed

Reviewer #2: All comments have been addressed

2. Is the manuscript technically sound, and do the data support the conclusions?

Reviewer #1: Yes

Reviewer #2: Yes

3. Has the statistical analysis been performed appropriately and rigorously? 

Reviewer #1: N/A

Reviewer #2: Yes

4. Have the authors made all data underlying the findings in their manuscript fully available?

Reviewer #1: Yes

Reviewer #2: Yes

5. Is the manuscript presented in an intelligible fashion and written in standard English?

Reviewer #1: Yes

Reviewer #2: Yes

6. Review Comments to the Author

Reviewer #1: The authors have replied sufficiently to all my comments. It is a very nice manuscript. Kazuki Sakakura

Reviewer #2: The authors have adequately addressed all of the concerns given in the previous review and revised the manuscript accordingly. I have no further comments.

7. PLOS authors have the option to publish the peer review history of their article (what does this mean?). If published, this will include your full peer review and any attached files.

Reviewer #1: **Yes: **Kazuki Sakakura

Reviewer #2: **Yes: **Masaki Izumi

---

## [Editor Report · Acceptance letter]

19 Jul 2024

PONE-D-24-17382R1 

PLOS ONE

Dear Dr. Devergnas, 

I'm pleased to inform you that your manuscript has been deemed suitable for publication in PLOS ONE. Congratulations! Your manuscript is now being handed over to our production team.

Kind regards, 

on behalf of

Dr. Ayataka Fujimoto 

Academic Editor

PLOS ONE